

# Weather, snow, and streamflow data from four western juniper-dominated experimental catchments in southwestern Idaho, USA.

Kormos, Patrick R.[1], Marks, Danny G.[1], Pierson, Frederick B.[1], Williams, C. Jason[1], Hardegree, Stuart P.[1], Boehm, Alex R.[1], Havens, Scott C.[1], Andrew, Hedrick[1], Cram, Zane K.[1], and Svejcar, Tony J.[2]

[1]Northwest Watershed Research Center, USDA, Agricultural Research Service, 800 Park Blvd, Suite 105, Boise, ID 83712
[2]Range and Meadow Forage Management Research Unit, USDA, Agricultural Research Service, 67826-A, Highway 205, Burns, Oregon, 97720 USA

*Correspondence to:* Patrick Kormos patrick.kormos@ars.usda.gov

**Abstract.**

Weather, snow, stream, topographic, and vegetation data are presented from the South Mountain Experimental Catchments. This study site was established in 2007 as a collaborative, long-term research laboratory to address the impacts of western juniper encroachment and woodland treatments in the interior Great Basin region of the western USA. The data provide detailed information on the weather and hydrologic response from four highly instrumented catchments in the late stages of woodland encroachment in a sagebrush steppe landscape. Hourly data from six meteorologic stations and four weirs have been carefully processed, quality checked, and are serially complete. These data are ideal for hydrologic, ecosystem, and biogeochemical modeling. Data presented are publicly available from the USDA National Agricultural Library administered by the Agricultural Research Service (https://data.nal.usda.gov/dataset/data-weather-snow-and-streamflow-data-four-western-juniper-dominated-experimental-catchments, doi:10.15482/USDA.ADC/1254010).

## 1 Introduction

Across the interior western US, native Western Juniper (*Juniperus occidentalis* Hook.) is encroaching into sagebrush-dominated (*Artemisia* spp.) landscapes. Because of the associated impacts on the ecosystem quality and local economy, juniper encroachment has become a critical issue to the region's resource managers and ranchers. Fire-sensitive native conifers in the western U.S. have greatly expanded in response to changing fire regimes following European settlement (Miller and Wigand, 1994; Miller and Rose, 1995; Weisberg et al., 2007; Miller et al., 2000). Western Juniper now dominates over 3.6 million ha of rangeland in the Intermountain Western US. Juniper (*Juniperus* spp.) expansion into sagebrush ecosystems influences the vegetation community (Bates et al., 2000; Miller et al., 2005; Miller and Tausch, 2001) and the hydrology and soil resources of an area (Pierson et al., 2007, 2010; Williams et al., 2014), which affect wildlife habitat. At mid to high elevations, expansion of native conifer species is viewed as a major threat to sagebrush obligates such as the greater sage grouse (*Centrocercus urophasianus*) (Braun, 1998; Connelly and Braun, 1997). Although the deleterious impact of juniper encroachment is widely reported through



field studies, there are limited datasets available to quantify that impact on larger scales through modeling. To address the need for monitoring data, the South Mountain Experimental Catchments were established in 2007 in a juniper-dominated region of southwestern Idaho, USA (Kormos et al., in press).

In this paper we present hourly weather, precipitation, snow, and streamflow data, along with lidar-derived topographic and vegetation cover, that detail the hydrologic function of a Western Juniper-dominated (*Juniperus occidentalis* Hook.) study area. Table 1 summarizes the hydro-meteorological variables presented in this paper with the instrument used to collect the data and instrument height. These data represent a relatively complete background hydrologic dataset that has been collected from 1 October 2007 through 30 September 2013 (six water years, WY2008 to WY2013), and are appropriate to force and evaluate models that investigate the hydrologic function and change in these systems. This time period is sufficient to provide a range of precipitation and temperature conditions typical for this region. A management plan that systematically removes juniper from one catchment at a time has been implemented effective in 2015.

## 2   Site Description

The South Mountain Experimental Catchments are located on South Mountain ($-116.90°W, 42.67°N$) in the Owyhee Mountains just east of the Idaho-Oregon border, in the northwestern USA (Fig. 1). The research catchments were established in 2007 as a collaborative, long-term research laboratory to assess the hydrologic and ecologic impacts of juniper encroachment and removal in the Great Basin region. Four west-draining catchments are defined by the locations of drop box weirs (Bonta and Pierson, 2003). Contributing areas range in size from 20.0 to 70.2 ha for a total of 204.5 ha (Table 2). Elevation ranges from 1665 to 1898 meters above sea level and mean catchment slope ranges from 10° to 13°. Vegetation is typical of woodland-encroached sagebrush steppe ecosystems. Diminished understory consists of sparse shrubs, grasses, and forbs, while over story is exclusively Western Juniper. Juniper cover ranges from 31-42% based on a 1 m pixel classification where maximum vegetation height greater than 1.5 m is classified as juniper (Kormos et al., in press). Juniper density is approximately 288 stems ha$^{-1}$ with a mean height of 7.3 m (Sankey et al., 2013).

Mean water year precipitation from the six precipitation gauges was 627 mm for the 6 year dataset (Fig.2). The majority of precipitation occurs in the fall, winter, and spring, with little accumulation in June, July, and August (Fig. 3). A seasonal snow pack commonly accumulates in November and melts out in March and April. Six weather stations are arranged to capture the spatial variability in weather across the study area (Fig.1). To capture elevation gradients, three weather stations are located on ridges (designated with a 2 in the name) and three are located at lower catchment elevations (designated with a 1 in the name). All weather stations are equipped with identical instrumentation (Table 1). A snow courses is located within 30 m of each of the weather stations.



## 3   Spatial data: digital elevation and vegetation models

One meter bare earth elevation and vegetation height data were derived from an airborne lidar survey (Fig. 4) acquired in November 2007. The lidar point density was 7 points per square meter resulting in a vertical accuracy of approximately 3 cm. Processing of the lidar dataset to obtain the bare earth elevation and canopy height models was done using tools developed by the Boise Center Aerospace Laboratory (BCAL, 2016) as describe by Streutker and Glenn (2006). These data provide an accurate 1 m snapshot of bare earth elevation and mean and maximum vegetation height for each of the study catchments (Sankey et al., 2013). In addition we provide a 10 m digital elevation model obtained by aggregating elevation data from the 1 m dataset (Fig. 5). Similarly, we provided a 10 m maximum vegetation height dataset created by taking the mean of the 1 m maximum vegetation pixel heights contained in the 10 m pixels. These data provide an accurate 10 m snapshot of bare earth elevation and maximum vegetation height for each of the study catchments that can be utilized in modeling projects (Kormos et al., in press). Additional spatial data includes shapefiles of weather station and weir locations, the delineations of catchment boundaries, and an estimate of the locations of the ephemeral stream network. All geographic data are in the Universal Transverse Mercator projected coordinate system using zone 11 and 1983 North American Datum (UTM, zone 11, NAD83). Catchment delineations, stream channels, mean catchment slope, and elevations were derived directly from the 1 m bare earth digital elevation model.

## 4   Weather data

Measured weather data are typical of forcing variables required to run hydrologic models, and include air temperature ($^o$C), relative humidity (kPa kPa$^{-1}$), precipitation (mm), wind speed (m s$^{-1}$) and direction (degree), and incoming solar radiation (W m$^{-2}$). Vapor pressure and dew point temperature are calculated from air temperature and relative humidity using methods developed by Marks et al. (1999), described by Reba et al. (2011) and refined by Marks et al. (2013). Air temperature, vapor pressure, relative humidity, and precipitation from all stations were plotted together for every month to perform quality control. All weather data are hourly and have been cleaned, gap-filled, and with the exception of wind direction, serially complete for WY2008 to WY2013. Data gaps and bad or noisy values have been filled using the most appropriate of either linear interpolation, or multiple linear regression to nearby weather stations with the same measured variable. Raw data are also provided for all measured weather variables.

### 4.1   Precipitation

Shielded precipitation was measured at the six weather stations using 8 inch Belfort-type gauges with Alter Shields (Hanson et al., 2001). Precipitation was filtered following Nayak et al. (2008) and wind corrected using the World Meteorological Organization protocol as described in Dingman (2002). Most of the annual precipitation falls in the cold winter season with dew point temperatures close to zero (Fig. 3d). This creates a dynamic precipitation regime, where some years accumulate substantial snowpacks, and some years accumulate very little snow (Fig. 3b). Precipitation phase was computed using methods



described by Marks et al. (2013). Though precipitation across the South Mountain Experimental Catchments is typically a mix of rain and snow, the region is snow-dominated, with 53% to 76% of water year precipitation falling as snow or mixed phase events (Fig. 2). The six year water year average is 627 mm (314 mm snow), with WY2011 being the wettest year with 867 mm (354 mm snow) and WY2012 being the driest year with 445 mm (202 mm snow). An example of the hourly cumulative
precipitation, divided into phase from weather station E2, is shown in Figure 3a.

### 4.2 Air temperature and humidity

Air temperature and relative humidity were measured at the six weather stations. Vapor pressure was calculated from air temperature and relative humidity, which was converted to dew point temperature using methods described by Marks et al. (2013). Average water year air temperature over the South Mountain research catchments for the six water years of this study
is 7.0°C, with WY2010 being the coldest (6.0°C) and WY2012 being the warmest (8.0°C). The mean water year dew point temperature was -3.0°C, with WY2011 being the most humid and also the wettest (-1.9°C) and WY2012 being the least humid and the driest (-3.7°C) of the six water years in this study. Average water year air temperature during storms for the six water years of this study is 0.5°C, with WY2008 being the coldest (-1.0°C) and WY2012 being the warmest (1.6°C). The mean water year dew point temperature during storms was -1.0°C, with WY2008 having the greatest percent snow (-2.6°C) and WY2011
having the least percent snow (-0.3°C). An example of mean monthly air and dew point temperatures, with the monthly range from weather station E2 is shown in Figure 3d.

### 4.3 Wind speed and direction

Wind speed and direction are measured at the six weather stations. The three low elevation sites (M1, F1 & G1) are sheltered by topography and vegetation, while the ridge-top sites (E2, M2 & G2) are wind-exposed. F1 is extremely wind sheltered by
both topography and vegetation with a mean wind speed of 0.7 m s$^{-1}$, while M2 is the the most wind-exposed with a mean wind speed of 2.4 m s$^{-1}$. The prevailing wind direction during precipitation is from the west (274°). The maximum wind speed recorded during six water years of the dataset of 14.3 m s$^{-1}$. We did not attempt to gap fill missing or bad data from the wind direction time series. An example of monthly mean wind speed and the monthly range of wind speed values from weather station E2 is shown in Figure 3c.

### 4.4 Incoming solar radiation

Incoming solar radiation is measured at the six weather stations. Solar radiation measurements from weather stations F1 and M1 are vegetation-affected in the mornings and evenings. The average solar loading at the F1 site was 12.9 MJ day$^{-1}$ m$^{-2}$, while at site E2 it was 16.1 MJ day$^{-1}$ m$^{-2}$. An example of monthly solar loadings from weather station E2 is shown in Figure 3c.



## 5 Snow and streamflow data

### 5.1 Snow data

Snow depth is continuously measured at the six weather stations. Because these automated snow depth measurements are inherently noisy, the data are processed using multiple smoothing windows. This practice allows for the cleaning of instrument noise, while maintaining sharp accumulation and melt events. We did not attempt to fill large time periods with excessively noisy data in the cleaned snow depth data file, and have denoted them as missing data (Fig. 3b, WY2011). Raw snow depth data are provided. In addition to automated snow depth measurements, manual measurements of snow water equivalent (SWE) were made two to three times each year at snow courses near the six weather stations using a federal-type snow tube. Snow courses were visited 16 times during the 6 year dataset, and were not measured in WY2009. These snow water equivalent values are reported in the final data file, and depths and densities are reported in the raw data file. An example of the cleaned snow depth from weather station E2 is shown in Figure 3b.

### 5.2 Streamflow

Stream stage is measured with Druck pressure transducers in stilling wells at four drop box v-notch weirs. Stage is converted to discharge with well-established rating curves (Bonta and Pierson, 2003). The streams that drain the South Mountain Experimental Catchments are intermittent and initiate in response to rain on snow or snow melt events (Fig. 3a and b). Streamflow ceases in late spring to mid summer. Mean water year discharge from all catchments across years was 115 mm (Fig. 2). Catchment M, which has the smallest contributing area has the lowest mean annual discharge at 90 mm. Catchment F has the highest mean annual stream discharge at 145 mm. The lowest stream yields occurred in WY2013 and the highest stream yields occurred in WY2011. Runoff ratios are approximated for the four South Mountain Experimental Catchments by assuming that the mean of precipitation measured by gauges within each catchment represents the precipitation that fell in that catchment. Average catchment runoff ratios varied from 0.07 for M to 0.22 for F. An example of the streamflow data included in this dataset from weir E is shown in Figure 3b.

## 6 Example Data

We present data from a mid February storm in 2012 as an example of the dynamic weather that is described in this manuscript (Fig. 6). At the start of this storm, the air was cold and saturated resulting in snowfall and an accumulation of about 9.5 cm of snow depth (Fig. 6b and c). Wind was relatively calm and cloud cover lead to low incoming solar radiation at all weather stations (Fig. 6a). Snow depth increased until midday on 21 February when air and dew point temperatures rose above freezing and caused precipitation to change from snow to mixed precipitation, and then to rain (Fig. 6b and c). Snow melt, and rain lead to streamflow initiation from catchments F, G, and M, and an increase in flow at weir E (Fig. 6c). An additional rain on snow event occurred from 22 February, 17:00 to 19:00 leading to increased streamflow at all weirs. Clear skies and warming





temperatures caused increased flow from the smallest catchment M on 24 February. A small snow event occurred in the early morning of 25 February, which lead to an increase in snow depth.

## 7 Data Availability

All data presented in this paper are available from the National Agricultural Library website (doi:10.15482/USDA.ADC/1254010). Included is a readme file that contains a detailed description of data file contents, including header information and contact information for additional details. Additional weather and hydrologic response data for the South Mountain Experimental Catchments and the Reynolds Creek Experimental Watershed are available at ftp://ftp.nwrc.ars.usda.gov/publicdatabase/.

## 8 Summary

Data presented in this paper support ongoing research in a mountain environment that is relevant to both native ecosystems and local economy in the Great Basin region of the northwestern US. This region has experienced extensive woodland encroachment into sagebrush-dominated landscapes, which has become a critical issue regarding the regional economy and ecosystem health. The data are unique because they capture the complicated weather-snow-streamflow dynamics representative of a large portion of the juniper impacted western US. In addition, the data provided represent model forcing variables that are commonly required to conduct modeling studies of the hydrologic and environmental systems in the region. Spatial data are derived from a lidar dataset and represent the topography and vegetation of the South Mountain Experimental Catchments at a 1 m resolution. In all, six water years of gap-filled and serially complete hourly weather, snow depth, and streamflow data are presented from six weather stations and four weirs, which represent the environmental gradients present in the study area.

*Acknowledgements.* We thank the Lowry and Stanford families for cooperation and property access, the BLM Boise District and Owyhee Field Offices, and specifically Skip Nyman, Tony Runnels, John Wilford, and Barry Caldwell for field support and data collection, and Dr. Rupesh Shrestha for Lidar processing and the development of the canopy height models. This research was funded in part by Grant NA05OAR4601137 from the NOAA Earth System Research Laboratory Physical Sciences Division, the BLM Owyhee Uplands Pilot Project [ISU-BLM Agreement #DLA060249; and ARS-BLM Agreement #DLI050018], by the NSF Idaho EPSCoR Program, by the NSF under award number EPS-0814387 and CBET-0854553, by USDA-NRCS Water and Climate Center-Portland, Oregon (60-5362-4-003), by the NSF Reynolds Creek CZO Project (58-5832-4-004), and by USDA-ARS CRIS Snow and Hydrologic Processes in the Intermountain West (5362-13610-008-00D).





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

**Tables and Figures**



**Table 1.** Hydro-meteorological variable, type of instrument, and instrument height from the South Mountain Experimental Catchments. Locations are denoted with a WS for weather station or W for weir.

| Hydro-meteorological Variable | Instrument / Method | Instrument Height (m) |
|---|---|---|
| Precipitation (WS) | 8 inch Belfort-type gauge with Alter Shield | 3 |
| Wind Speed (WS) | Met One WS 013 | 3 |
| Wind Direction (WS) | Met One WD 023 | 3 |
| Air Temperature (WS) | Vaisala HMP45AC | 3 |
| Relative Humidity (WS) | Vaisala HMP45AC | 3 |
| Vapor Pressure (WS) | Calculated from Air Temperature and Relative Humidity | 3 |
| Dew Point Temperature (WS) | Marks et al (2013) | 3 |
| Incoming Solar Radiation (WS) | Kipp and Zonnen CMP3 | 3 |
| Snow Depth (WS) | Judd Ultrasonic Depth Sensor | 3 |
| Snow Course (WS) | Federal-Type Snow Tube | na |
| Stream Discharge (W) | Druck PDCR1830 in drop box v-notch weir | na |

**Table 2.** Watershed areas, the percent of the pixels classified as juniper dominated, elevation ranges, and mean slopes. Mean catchment elevations are in parentheses.

| Watershed | Area (ha) | Juniper Cover (%) | Elevation Range (m) | Mean Slope (degree) |
|---|---|---|---|---|
| E | 56.7 | 42 | 1704-1898 (1793) | 13 |
| F | 56.6 | 61 | 1687-1815 (1748) | 13 |
| G | 70.2 | 53 | 1693-1814 (1758) | 12 |
| M | 21.0 | 54 | 1665-1791 (1723) | 10 |




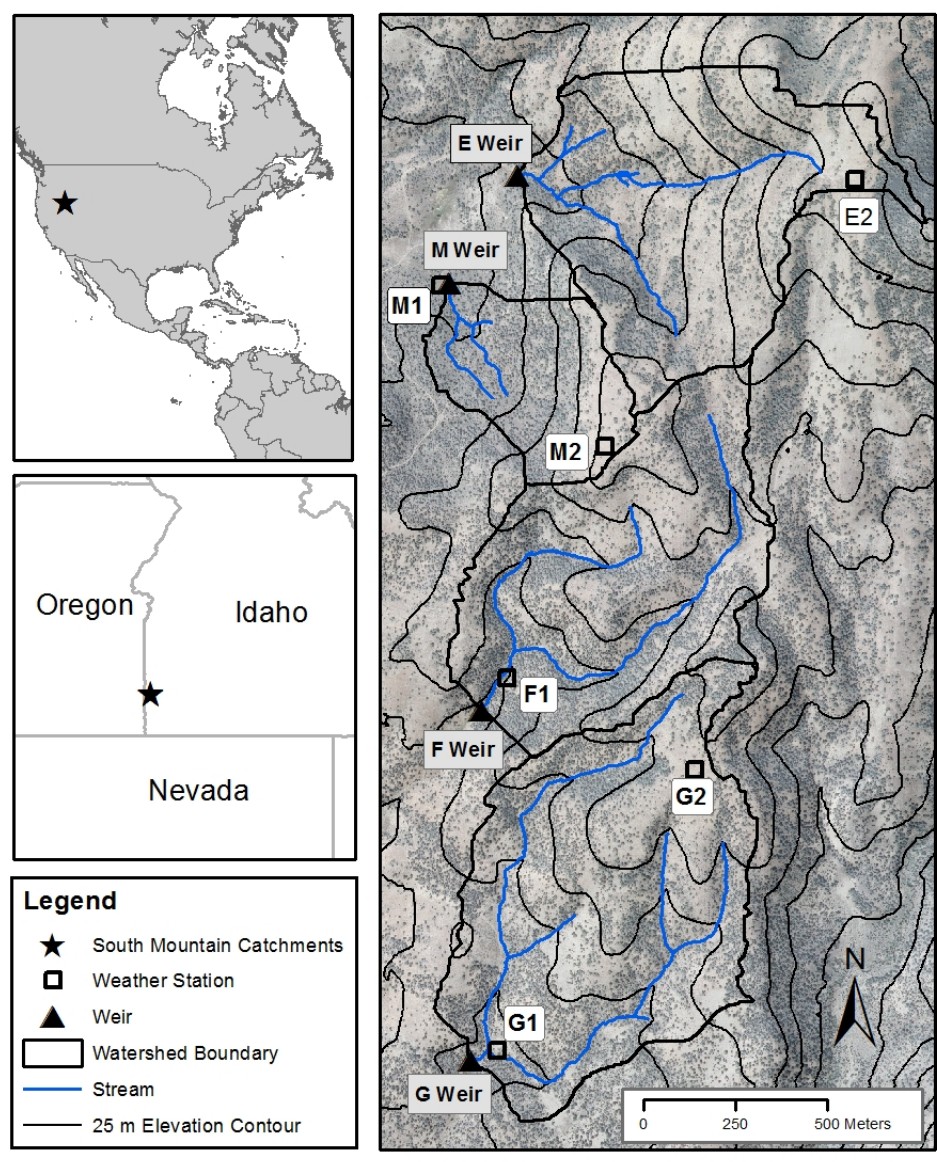

**Figure 1.** Location map of the South Mountain Experimental Catchments showing the locations of weather stations and weirs. The base map shows the distribution of Western Juniper.



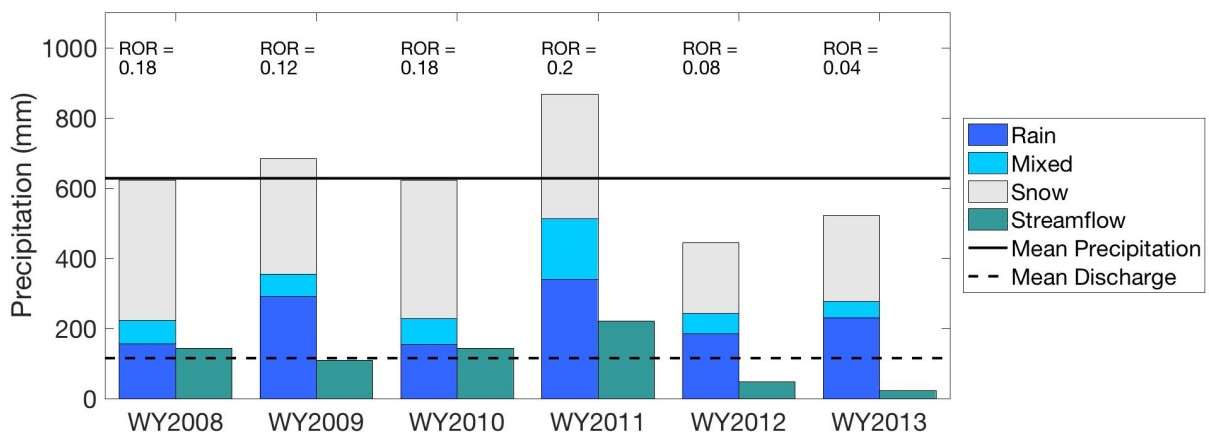

**Figure 2.** Total water year precipitation split up into phase at the South Mountain Experimental Catchments showing that the precipitation regime is snow dominated. Mean wateryear precipitation was 627 mm as depicted by the solid black line. Mean total catchment streamflow is shown as green bars. The 6-year mean water year stream flow was 115 mm as shown with the dashed black line. Runoff ratios (ROR) are displayed for each water year above the bars.

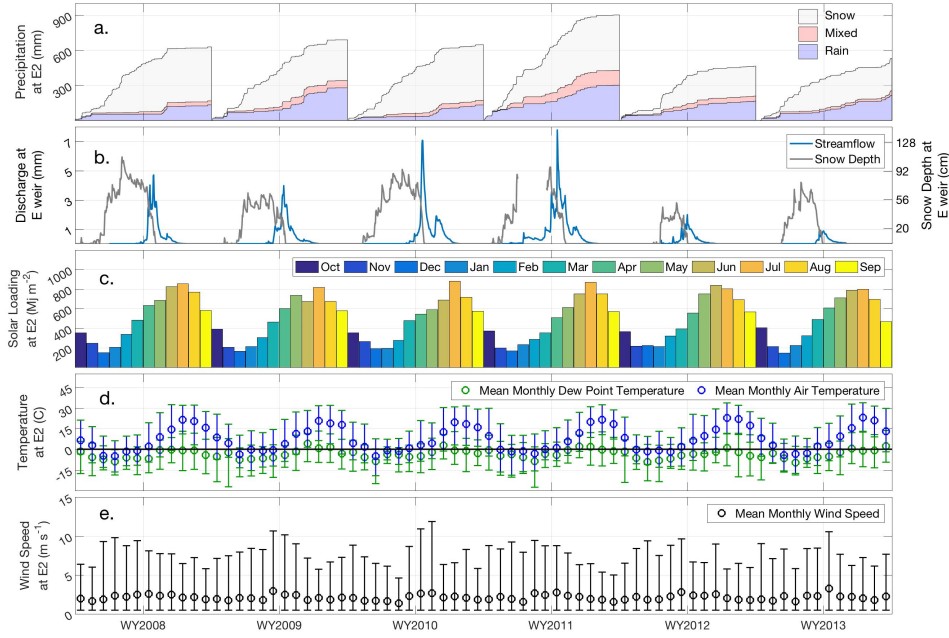

**Figure 3.** Example forcing data from Catchment E and Weather Station E2 showing a) precipitation amount and phase, b) streamflow, c) monthly incoming solar radiation, d) air and dew point temperature, and e) wind speed.



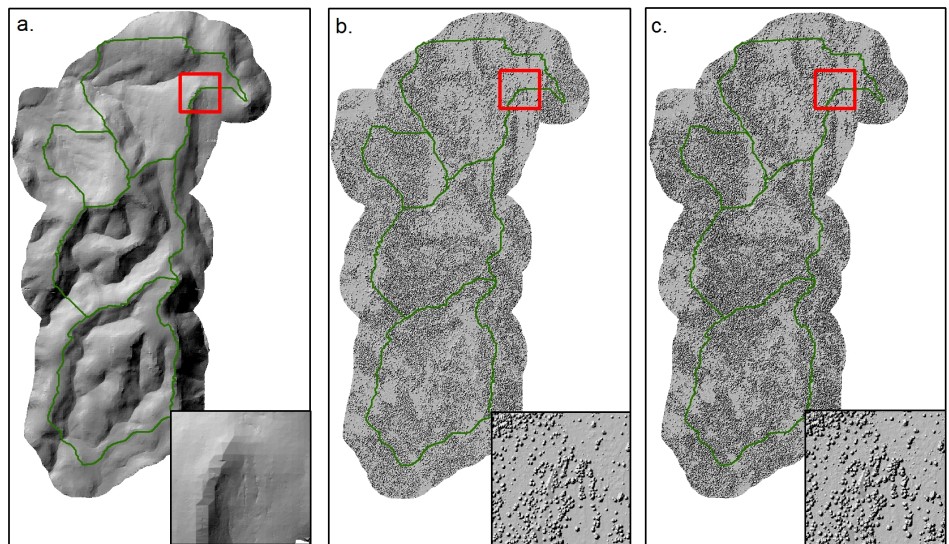

**Figure 4.** Hillshades of 1 m bare earth model, 1 m mean vegetation heights, and 1 m maximum vegetation heights. The insets in the lower right hand corners show close up images of the area shown in the red boxes

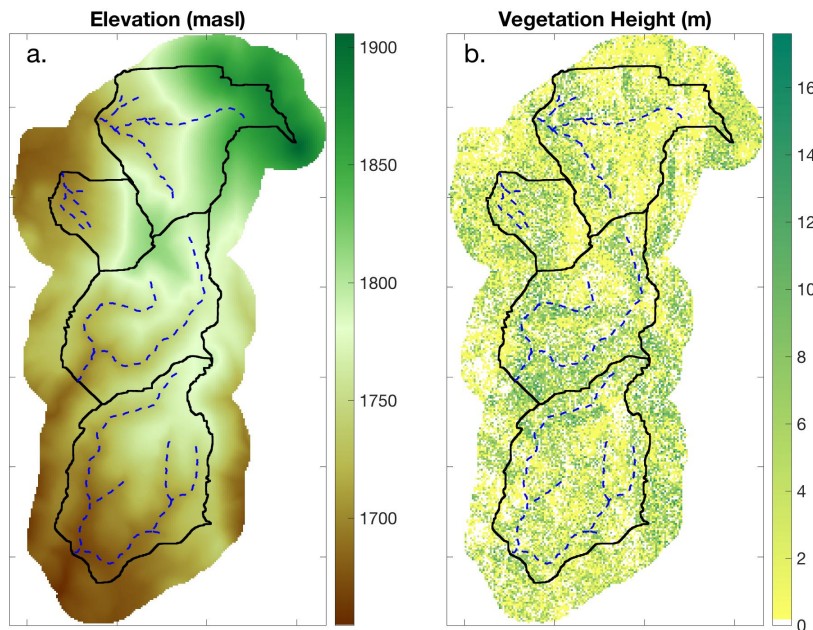

**Figure 5.** a) Ten meter elevation map of the South Mountain Experimental Catchments showing the catchment boundaries and stream locations and b) ten meter vegetation height map.




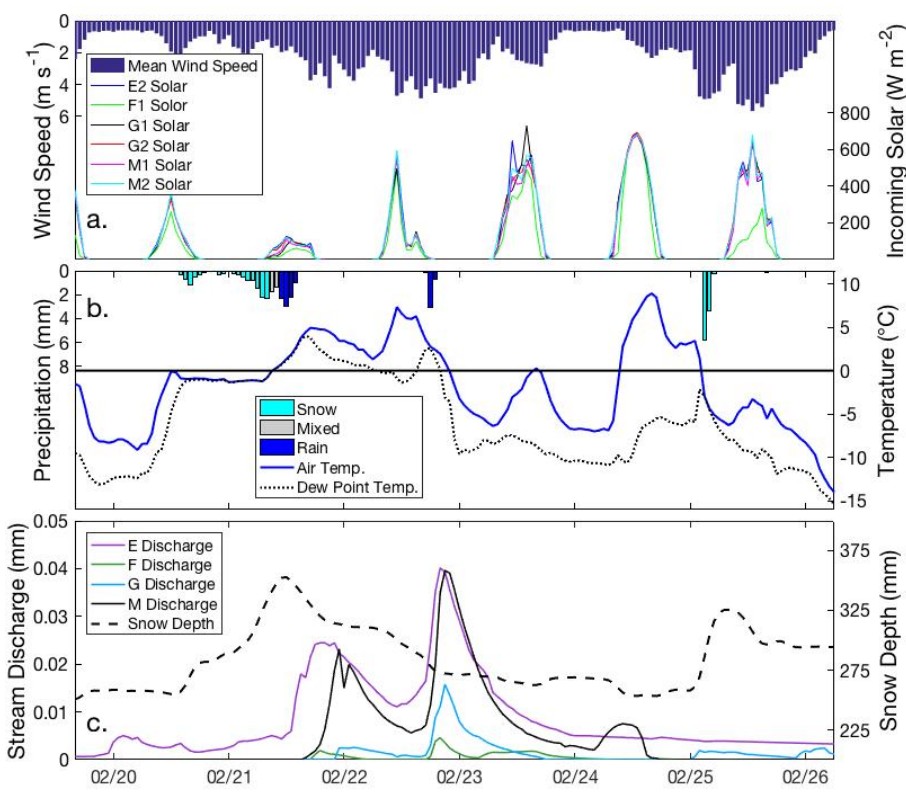

**Figure 6.** Example weather and response data from a January storm at the South Mountain Experimental Catchments showing a. mean precipitation mass and phase with air and dew point temperatures, b. mean wind speed and incoming solar radiation, and c. streamflow and snow depth response data.