# Peer review of "Weather, snow, and streamflow data from four western juniper-dominated experimental catchments in southwestern Idaho, USA."

_Earth System Science Data, 2016_

## Referee Comment (RC1) · Anonymous Referee #1 · 11 Oct 2016

General comments: Authors are to be commended for submitting a polished manuscript. The data set is comprehensive and can be used to study mountain hydrology in semi-arid catchments. The data may be used to drive physics-based snow or/and hydrology models. However, before the manuscript can be accepted in as is form, authors are encouraged to address the following comments:

Specific comments:

– The https link in abstract does not take me to the data. One has to search for the relevant data on nal.usda.gov. I wonder if this can corrected. However, the doi link

worked just fine.

– My experiences with accessing the data set: * When i clicked "readme" file, it showed "The requested URL "/system/files/readme_2.txt" was not found on this server.". Please correct it. * The "measurement location coordinates" link showed me 10 locations. Are these for the snow course observations. Explanation of the data is missing.

– In the "Introduction" section, it is noted that western Juniper is encroaching into the sagebrush-dominated landscape in the interior Great Basin region, and the presented data will facilitate the study of the impacts of Juniper encroachment on ecohydrology. It is not clear if any of the discussed catchments present a base case with zero to minimal encroachment. If there is such a watershed, please identify it. If not, authors are encouraged to highlight sites that are "juniper influenced" vs. "sage influenced" that may allow understanding the effects of juniper encroachment. Alternatively, sufficient explanations should be provided on how the data sets may be used to study the impacts of juniper encroachment.

– P3,L31 and P4,L1: "Precipitation phase was computed using methods described by Marks et al., (2013)". Marks et al. compared 4 methods for estimating precipitation phase. I did not see the phase data from all four methods. If evaluation of the phase was done based on a certain method (e.g. dew-point temperature method) only, clearly state so.

– P3, L21: It is noted that wind direction time series was not filled, while other data sets were. Please add a short statement explaining why wind direction wasn't/can't be filled.

– P3, L29: The WMO protocol used in Dingman's book should be properly referenced by providing the page number. Otherwise, it is difficult to cross-check.

– P5, L7: Please provide more information on the thresholds or methods used to determine "excessively noisy data".

– P6, L7: The ftp://ftp.nwrc.ars.usda.gov/publicdatabase/ is very slow. Also, I did not see any "additional" data of south mountain on this website as has been claimed. If there is some, why has that been not added to the NAL website. The FTP site also does not appear to have the details of data. I suggest removing this link. It is also not clear why Reynolds Creek watershed is mentioned here.

– While the presented data set is rich, considering that it is designed to be used for snow and hydrologic modeling, there are some important variables that are missing. For example, most radiation or snow interception/melt/accumulation models use LAI or shape of vegetation (e.g. cylindrical canopy with given minimum/maximum height and diameter at certain height) as inputs. Given the LiDAR data, can the aforementioned variables (LAI or canopy shape) be generated/provided. The would significantly improve the usability of this data by snow and hydrology modeling. Kormos et al. (2016, RE&M) have used the presented data sets for modeling, but it is not clear how the LAI was derived.

Another important data that is often required by snow and hydrology modelers for validation is SWE. It appears that SWE is only available few times during the water year. A line or two highlighting this limitation and how the presented data set may still be used for validation, should be included in the text. If there is any data of soil properties from the catchments, authors are encouraged to include those in the data set.

– Some discussion of the juniper removal plan and which watersheds they are being implemented in, should be included in the summary. This will allow readers to identify watersheds of interests where one can study the impacts of juniper removal.

Minor concerns:

– P1,L19: Consider revising "which affect wildlife habitat" to "which in turn also affects the wildlife habitat". Provide a reference or two regarding the affects on wildlife habitat.

– P2,L1: "there are limited datasets available to quantify the impact on larger scales

through modeling". Provide references for "limited datasets" if there are any. Also, it is not clear if the previously published data sets are at points or at plot scales. That will give the reader some idea of what is meant by "larger scales" here. Is it the watershed scale or an area larger than a certain threshold.

– P2,L28: "A snow courses is" should be revised to "A snow course is"

---

## Referee Comment (RC2) · Anonymous Referee #2 · 18 Oct 2016

Review of 'Weather, snow, and streamflow data from four western juniper-dominated experimental catchments in southwestern Idaho, USA' by Patrick Kormos et al.

The authors present a six-year hydrometeorologic dataset from four neighboring juniper-dominated experimental catchments. Data are presented from six meteorological stations and four streamflow weirs. Also included are lidar-derived DEM and vegetation height models. The datasets are of excellent quality and provide the necessary input and verification data for hydrologic simulations. The paper is well-written and data are well-described. I find no major flaws and have only limited minor comments.

[Figure]

In my opinion, the paper and dataset are publishable with adequate attention to these points.

Page 1, Line 1: The authors should be more clear in the description of the data being published. "Weather, snow, stream, topographic, and vegetation data . . ." should be clarified as "Meteorological, snow, streamflow, topographic, and vegetation height data . . .". For example, 'stream data' is vague and could be interpreted differently by a hydrologist, geomorphologist, or biogeochemist. The vegetation data is limited to height data. Best to be as clear as possible in this first sentence.

Page 1, Line 13: The logical order of the first paragraph could be improved.

Page 1, Line 13: The sentence starting with 'Because' doesn't adequately describe the issues facing managers and ranchers w.r.t. juniper encroachment, in my opinion. Please provide a succinct example of a specific challenge that encroachment presents to each group, rather than a general statement (ecological and economic impacts) that isn't elaborated upon. E.g., how juniper encroachment economically impacts ranchers is not explained.

Page 1, Line 15: If the 'changing fire regimes' term describes 'fire suppression efforts', please state that.

Page 1, Line 17: Move the Juniperus spp. definition to the first use of the word 'juniper' on Line 13.

Page 2 'Site Description': I think the fact that the catchments are neighboring (many share borders) is a unique characteristic that should be described. For example, some distributed hydrological models may benefit from this information in the treatment of lateral connectivity.

Page 2 'Site Description': Please consider providing a size metric for each lidar product (e.g., the # of grid cells in the east and west directions).

Page 2 'Site Description': Please describe the buffer distance around the catchment

boundaries (i.e., that the lidar products are not tightly 'cropped' to the catchment extent).

Page 2, Line 13: I am accustomed to the order (latitude, longitude) rather than the reverse.

Page 2, Line 23: I prefer spelling out 'six-year' rather than '6 year'. Here and elsewhere.

Page 3, Line 1: " . . . a snow-free airborne lidar survey . . ."?

Page 3, Line 5: typo.: 'described'

Page 3, Line 30: Change 'zero' to 'the freezing point' or to 0°C.

Page 4, Lines 7-9: The second sentence is largely redundant with the first paragraph of this section. I suggest: "Dew point temperature was calculated from measured values of air temperature and relative humidity (Marks et al., 2013)."

Page 4, Line 22: Typo: change ". . . of the dataset of 14.3 . . ." to ". . . of the dataset was 14.3 . . ."

Page 6, Line 1: 'Catchment M' should have a capital 'C'

Page 6, Line 15: change 'at a 1 m resolution' to 'at 1 m resolution'.

Page 6, Lines 16-17: Suggest changing 'represent' to 'adequately capture'.

Figure 1: Label one upper and one lower contour line to give the reader a better sense of the elevation distribution. Please state the contour interval in the figure caption.

Figure 6: I think this should be a February storm event (typo. in caption that says 'January').
* * *

---

## Author Comment (AC1) · 24 Oct 2016

**Author responses to the RC1 comments on Manuscript: ESSD-2016-42, entitled " Weather, snow, and streamflow data from four western juniper-dominated experimental catchments in southwestern Idaho, USA" – Earth System Science Data**

- **Reviewer comments are in bold font and author responses are in normal font.**
- **Line number references are to the original manuscript unless otherwise noted.**
- **Quotes from the text are italicized and proposed revisions are underlined.**

*RC1 General Comments:*

**Authors are to be commended for submitting a polished manuscript. The data set is comprehensive and can be used to study mountain hydrology in semi-arid catchments. The data may be used to drive physics-based snow or/and hydrology models. However, before the manuscript can be accepted in as is form, authors are encouraged to address the following comments:**

We thank the reviewer for a thorough review and have addressed the comments below.

*RC1 Specific Comments:*

1. **The https link in abstract does not take me to the data. One has to search for the relevant data on nal.usda.gov. I wonder if this can corrected. However, the doi link worked just fine.**

   The link works for me if copied and pasted into my browser on my desktop, but had some problems doing the same thing on my laptop. I double-checked that it is the correct address. I don't quite know how to fix this problem. I will ask the editor and the people that type set the article to try to address this issue. Thank you for bringing this up.

2. **My experiences with accessing the data set: * When i clicked "readme" file, it showed "The requested URL "/system/files/readme_2.txt" was not found on this server.". Please correct it. * The "measurement location coordinates" link showed me 10 locations. Are these for the snow course observations. Explanation of the data is missing.**

   Thanks for pointing this out. I have worked with the National Ag. Library to fix this issue.

The measurement location coordinates file has the locations of where the weather stations and weirs are located. I have modified the file description as follows in both the readme.txt file and the link to the station_coords.csv file: "*Station coordinates and elevations of weirs (sme, smf, smg, smm) and weather stations (sme2, smf1, smg1, smg2, smm1, smm2) in meters. Coordinates were measured using a Garmin hand held GPS with approximately 3 m accuracy. Elevations are obtained from a 1 meter Liar-derived digital elevation model corresponding to the coordinates. See above for spatial reference information.*"

3. **In the "Introduction" section, it is noted that western Juniper is encroaching into the sagebrush-dominated landscape in the interior Great Basin region, and the presented data will facilitate the study of the impacts of Juniper encroachment on ecohydrology. It is not clear if any of the discussed catchments present a base case with zero to minimal encroachment. If there is such a watershed, please identify it. If not, authors are encouraged to highlight sites that are "juniper influenced" vs. "sage influenced" that may allow understanding the effects of juniper encroachment. Alternatively, sufficient explanations should be provided on how the data sets may be used to study the impacts of juniper encroachment.**

All catchments are in phase III encroachment (fully encroached). Impact study may occur after treatment of catchments. We have included a description of that plan on page 2, line 3 as follows: "*Catchment M was burned in the fall of 2015 and catchment G is scheduled to burn in the spring of 2017. The long term treatment plan includes burning catchments F and then E.*" In addition, we have removed the sentence on page 2, line 10 that also described the treatment plan in less detail.

We have indicated that this dataset includes pretreatment data on page 2, line 4 as follows: "*In this paper we present hourly pretreatment weather, precipitation…*".

We have also restructured the last paragraph of the introduction to include an explanation/example of how the dataset can be used now to study the impact of juniper encroachment (Page 2, line 7): "*These data represent a relatively complete background hydrologic dataset that has been collected from 1 October 2007 through 30 September 2013 (six water years, WY2008 to WY2013). This time period is sufficient to provide a range of precipitation and temperature conditions typical for this region. These data and are appropriate to force and evaluate models that investigate the hydrologic function and change in these systems. For example, Kormos et al., (in press) utilized this dataset to evaluate the changes in ecosystem water availability between juniper-dominated and*

*sagebrush-dominated landscapes by simulating snow dynamics with and without juniper trees."*

4. **P3, L31 and P4,L1: "Precipitation phase was computed using methods described by Marks et al., (2013)". Marks et al. compared 4 methods for estimating precipitation phase. I did not see the phase data from all four methods. If evaluation of the phase was done based on a certain method (e.g. dew-point temperature method) only, clearly state so.**

   We have changed this sentence to "*Precipitation phase was computed using the dew point temperature methods as described by Marks et al. (2013)."*

5. **P3, L21 (P4, L22): It is noted that wind direction time series was not filled, while other data sets were. Please add a short statement explaining why wind direction wasn't/can't be filled.**

   We have included additional verbiage describing wind direction data on page 4, line 22 as follows: "*We did not attempt to gap fill missing or bad data from the wind direction time series, as correlations between wind measurement stations are low. However, there is sufficient wind direction data to obtain average wind directions during water years and individual storms.*"

6. **P3, L29: The WMO protocol used in Dingman's book should be properly referenced by providing the page number. Otherwise, it is difficult to cross-check.**

   We have included the page number on page 3, line 29 as suggested.

7. **P5, L7: Please provide more information on the thresholds or methods used to determine "excessively noisy data".**

   We have added a sentence on page 5, line 6 to explain excessively noisy data as follows: "*Excessively noisy data was identified as time periods that contained more erroneous measurements than reasonable measurements.*"

8. **P6, L7: The ftp://ftp.nwrc.ars.usda.gov/publicdatabase/ is very slow. Also, I did not see any "additional" data of south mountain on this website as has been claimed. If there is some, why has that been not added to the NAL website. The FTP site also does not appear to have the details of data. I suggest removing this link. It is also not clear why Reynolds Creek watershed is mentioned here.**

   We have removed the reference to Reynolds Creek as suggested. We have left the link to additional data as the ftp site has the level 1 data (obvious erroneous values flagged and removed) from earlier and later than this data

set. This data is "raw" data and has not gone through the quality control that WY2008 – WY2013 has been subjected to. In addition, this database will be updated in the future with more current weather station and streamflow data. The speed of our server is expected to improve soon, as well.

9. **While the presented data set is rich, considering that it is designed to be used for snow and hydrologic modeling, there are some important variables that are missing. For example, most radiation or snow interception/melt/accumulation models use LAI or shape of vegetation (e.g. cylindrical canopy with given minimum/maximum height and diameter at certain height) as inputs. Given the LiDAR data, can the aforementioned variables (LAI or canopy shape) be generated/provided. The would significantly improve the usability of this data by snow and hydrology modeling. Kormos et al. (2016, RE&M) have used the presented data sets for modeling, but it is not clear how the LAI was derived.**

   The Kormos et al. (2016) paper did not use LAI or canopy shape in the modeling of the South Mountain Watersheds. It used mean and maximum canopy height on a 10m grid. The raw lidar point cloud is available for those that need more specific data sets from: https://www.idaholidar.org/data/data-map/south-mountain/ We have included this link on page 3, line 11 as follows: "*The raw lidar point cloud is available through Idaho Lidar Consortium (https://www.idaholidar.org/data/data-map/south-mountain/) in the case that additional spatial data is required, such as LAI or vegetation shape parameters.*"

10. **Another important data that is often required by snow and hydrology modelers for validation is SWE. It appears that SWE is only available few times during the water year. A line or two highlighting this limitation and how the presented data set may still be used for validation, should be included in the text. If there is any data of soil properties from the catchments, authors are encouraged to include those in the data set.**

   We have included the following verbiage on page 5, line 9 pointing out this limitation as follows: "*Although significant resources were expended collecting SWE data, we recognize that this is a limited model validation dataset. The combination of continuous snow depth and SWE measurements should be sufficient to evaluate distributed snow model results.*"

   Although we agree that it would be great to have measured soil properties for this catchment, there is no quantitative soil property data from South Mountain, and users would have to rely on national soil databases.

11. **Some discussion of the juniper removal plan and which watersheds they are being implemented in, should be included in the summary. This will allow**

**readers to identify watersheds of interests where one can study the impacts of juniper removal.**

We have included the following sentences on page 6, line 12 to describe the treatment schedule as follows: "*This publication provides details on background data from catchments that are now juniper dominated. A treatment schedule to remove juniper is now being implemented so comparative studies can be conducted. Catchment M was burned in the fall of 2015 and catchment G is scheduled to burn in the spring of 2017. Catchments F and E are also to be treated.*"

*RC1 Minor Concerns:*
1. **P1,L19: Consider revising "which affect wildlife habitat" to "which in turn also affects the wildlife habitat". Provide a reference or two regarding the affects on wildlife habitat.**

   We have made the correction as suggested.

2. **P2,L1: "there are limited datasets available to quantify the impact on larger scales through modeling". Provide references for "limited datasets" if there are any. Also, it is not clear if the previously published data sets are at points or at plot scales. That will give the reader some idea of what is meant by "larger scales" here. Is it the watershed scale or an area larger than a certain threshold.**

   To our knowledge, there is no openly available datasets. We use the limited verbiage because we know those data exist, but you have to collaborate to get ahold of it. For example, SageSTEP has an extensive database of vegetation conditions and changes due to treatment that you might be able to use if you collaborate through the following process.

   SageSTEP: Opportunities for Additional Studies, c2005-2013, Union, Oregon: Sagebrush Steppe Treatment Evaluation Project; [accessed 2016 Oct 13]. http://www.sagestep.org/collaborative_projects/opportunities.html.

   In addition, Camp Creek in Oregon has collected juniper and hydrology data, but this data is not freely available or published as far as I can find.

3. **P2,L28: "A snow courses is" should be revised to "A snow course is"**

   We have made this change as suggested.

---

## Author Comment (AC2) · 10 Nov 2016

**Author responses to the RC2 comments on Manuscript: ESSD-2016-42, entitled " Weather, snow, and streamflow data from four western juniper-dominated experimental catchments in southwestern Idaho, USA" – Earth System Science Data**

- Reviewer comments are in bold font and author responses are in normal font.
- Line number references are to the original manuscript unless otherwise noted.
- Quotes from the text are italicized and proposed revisions are underlined.

*RC2 General Comments:*

**The authors present a six-year hydrometeorologic dataset from four neighboring juniper-dominated experimental catchments. Data are presented from six meteorological stations and four streamflow weirs. Also included are lidar-derived DEM and vegetation height models. The datasets are of excellent quality and provide the necessary input and verification data for hydrologic simulations. The paper is well-written and data are well-described. I find no major flaws and have only limited minor comments. In my opinion, the paper and dataset are publishable with adequate attention to these points.**

We thank the reviewer for a thorough review and have addressed the comments below.

*RC2 Specific Comments:*

1. **Page 1, Line 1: The authors should be more clear in the description of the data being published. "Weather, snow, stream, topographic, and vegetation data …" should be clarified as "Meteorological, snow, streamflow, topographic, and vegetation height data . . .". For example, 'stream data' is vague and could be interpreted differently by a hydrologist, geomorphologist, or biogeochemist. The vegetation data is limited to height data. Best to be as clear as possible in this first sentence.**

   We have made the change as suggested here and in the title. Thank you.

2. **Page 1, Line 13: The logical order of the first paragraph could be improved.**

   We have reordered the first few paragraphs to try to improve the logical flow of information as follows:

*"Across the interior western US, native Western Juniper (Juniperus occidentalis Hook.) is encroaching into sagebrush-dominated (Artemisia spp.) landscapes. These fire-sensitive native conifers in the western U.S. have greatly expanded in response to changing fire regimes following European settlement (Miller and Wigand, 1994; Miller and Rose, 1995; Weisberg et al., 2007; Miller et al., 2000). Western Juniper now dominates over 3.6 million ha of rangeland in the Intermountain Western US. Juniper (Juniperus spp.) expansion into sagebrush ecosystems influences the vegetation community (Bates et al., 2000; Miller et al., 2005; Miller and Tausch, 2001) and the hydrology and soil resources of an area (Pierson et al., 2007, 2010; Williams et al., 2014), which in turn also affects the wildlife habitat. For example, research in similar study sites demonstrate that juniper encroachment diminishes understory biomass (Bates et al., 2000, 2014; Pierson et al., 2013), which serves as a soil stabilization mechanism, forage for livestock, and habitat diversity. At mid to high elevations, expansion of native conifer species is viewed as a major threat to sagebrush obligates such as the greater sage grouse (Centrocercus urophasianus) (Braun, 1998; Connelly and Braun, 1997). Because of the associated impacts on the ecosystem quality and local economy (Aldrich et al., 2005), juniper encroachment has become a critical issue to the region's resource managers and ranchers.*

*Although the deleterious impact of juniper encroachment is widely reported through field studies, there are limited datasets available to quantify that impact on larger scales through modeling. To address the need for monitoring data, the South Mountain Experimental Catchments were established in 2007 in a juniper-dominated region of southwestern Idaho, USA (Kormos et al., in press). A period of background data collection has spanning the 2008-2015 water years. The catchments are now being treated to remove juniper so comparative studies can be conducted. Catchment M was burned in the fall of 2015 and catchment G is scheduled to burn in the spring of 2017. The long term treatment plan includes burning catchments F and then E."*

3. **Page 1, Line 13: The sentence starting with 'Because' doesn't adequately describe the issues facing managers and ranchers w.r.t. juniper encroachment, in my opinion. Please provide a succinct example of a specific challenge that encroachment presents to each group, rather than a general statement (ecological and economic impacts) that isn't elaborated upon. E.g., how juniper encroachment economically impacts ranchers is not explained.**

We have included the following sentence in the first paragraph to provide an example of the juniper encroachment challenges. Please see our response to the previous comment to see this sentence in the context of the paragraph.

*"For example, research in similar study sites demonstrate that juniper encroachment diminishes understory biomass (Bates et al., 2000, 2014; Pierson*

*et al., 2013), which serves as a soil stabilization mechanism, forage for livestock, and habitat diversity."*

And included the following citation in the sentence on economic impacts:

*Because of the associated impacts on the ecosystem quality and local economy (Aldrich et al., 2005), juniper encroachment has become a critical issue to the region's resource managers and ranchers.*

4. **Page 1, Line 15: If the 'changing fire regimes' term describes 'fire suppression efforts', please state that.**

   We have made the change as suggested as follows:

   "*These fire-sensitive native conifers in the western U.S. have greatly expanded in response to changing fire regimes (increased woody fuels in response to fire suppression efforts) following European settlement (Miller and Wigand, 1994; Miller and Rose, 1995; Weisberg et al., 2007; Miller et al., 2000).*"

5. **Page 1, Line 17: Move the Juniperus spp. definition to the first use of the word 'juniper' on Line 13.**

   This has been addressed in the reordering of the first paragraph as suggested. Please see the response to comment 2 above.

6. **Page 2 'Site Description': I think the fact that the catchments are neighboring (many share borders) is a unique characteristic that should be described. For example, some distributed hydrological models may benefit from this information in the treatment of lateral connectivity.**

   We have included the following sentence as suggested.

   "*Four west-draining catchments are defined by the locations of drop box weirs (Bonta and Pierson, 2003). The catchments share one or two borders with each other, which may be beneficial to hydrologic modeling efforts to describe lateral connectivity of basins or woodland treatment impacts beyond watershed divides. Contributing areas 25 range in size from 20.0 to 70.2 ha for a total of 204.5 ha (Table 2).*"

7. **Page 2 'Site Description': Please consider providing a size metric for each lidar product (e.g., the # of grid cells in the east and west directions).**

   We have added the size as suggested as follows on page 3, line 6:

*"These data provide an accurate 1 m snapshot (3276 rows and 1754 columns, 5,746,104 pixels with data) of bare earth elevation and mean and maximum vegetation height for each of the study catchments (Sankey et al., 2013)."*

And on page 3, line 9 as follows:

*"These data provide an accurate 10 m snapshot (329 rows and 176 columns, 37,310 pixels with data) of bare earth elevation and maximum vegetation height for each of the study catchments that can be utilized in modeling projects (Kormos et al., in press)."*

8. **Page 2 'Site Description': Please describe the buffer distance around the catchment boundaries (i.e., that the lidar products are not tightly 'cropped' to the catchment extent).**

   We have added the description as suggested on page 3, line 4 as follows:

   "*The lidar dataset extends beyond the catchment boundaries by approximately 200 m in most cases, although improved catchment boundaries extend to the end of the dataset in the northwest of the study area.*"

9. **Page 2, Line 13: I am accustomed to the order (latitude, longitude) rather than the reverse.**

   We have switched the order as suggested.

10. **Page 2, Line 23: I prefer spelling out 'six-year' rather than '6 year'. Here and elsewhere.**

    We have made these changes throughout the manuscript.

11. **Page 3, Line 1: " . . . a snow-free airborne lidar survey . . ."?**

    We have added "*snow-free*" as suggested.

12. **Page 3, Line 5: typo.: 'described'**

    We have fixed this typo. Thank you.

13. **Page 3, Line 30: Change 'zero' to 'the freezing point' or to 0°C.**

    We have made this change as suggested

14. **Page 4, Lines 7-9: The second sentence is largely redundant with the first paragraph of this section. I suggest: "Dew point temperature was calculated**

**from measured values of air temperature and relative humidity (Marks et al., 2013).”**

We have made this change as suggested. Thank you.

15. **Page 4, Line 22: Typo: change "... of the dataset of 14.3 ..." to "... of the dataset was 14.3 ..."**

We have corrected this typo as suggested.

16. **Page 6, Line 1: 'Catchment M' should have a capital 'C'**

We have made this correction.

17. **Page 6, Line 15: change 'at a 1 m resolution' to 'at 1 m resolution'.**

We have make this change as suggested.

18. **Page 6, Lines 16-17: Suggest changing 'represent' to 'adequately capture'.**

We have made this change as suggested

19. **Figure 1: Label one upper and one lower contour line to give the reader a better sense of the elevation distribution. Please state the contour interval in the figure caption.**

We have added one upper and one lower contour label as suggested and have added the following text to the figure caption:

"*The contour interval is 25 m, with the 1875 m and the 1725 m contours labeled for reference.*"

[Figure]

20. **Figure 6: I think this should be a February storm event (typo. in caption that says 'January').**

We have made this correction.

---

## Author Comment (AC3) · 10 Nov 2016

**Meteorological, snow,  streamflow, topographic, and vegetation height data from four western juniper-dominated experimental catchments in southwestern Idaho, USA.**

Kormos, Patrick R.[1], Marks, Danny G.[1], Pierson, Frederick B.[1], Williams, C. Jason[1], Hardegree, Stuart P.[1], Boehm, Alex R.[1], Havens, Scott C.[1], Hedrick, Andrew[1], Cram, Zane K.[1], and Svejcar, Tony J.[2]

[1]Northwest Watershed Research Center, USDA, Agricultural Research Service, 800 Park Blvd, Suite 105, Boise, ID 83712
[2]Range and Meadow Forage Management Research Unit, USDA, Agricultural Research Service, 67826-A, Highway 205, Burns, Oregon, 97720 USA

*Correspondence to:* Patrick Kormos patrick.kormos@ars.usda.gov

**Abstract.**

Meteorological, snow, streamflow, topographic, and vegetation height data are presented from the South Mountain Experimental Catchments. This study site was established in 2007 as a collaborative, long-term research laboratory to address the impacts of western juniper encroachment and woodland treatments in the interior Great Basin region of the western USA. The data provide detailed information on the weather and hydrologic response from four highly instrumented catchments in the late stages of woodland encroachment in a sagebrush steppe landscape. Hourly data from six meteorologic stations and four weirs have been carefully processed, quality checked, and are serially complete. These data are ideal for hydrologic, ecosystem, and biogeochemical modeling. Data presented are publicly available from the USDA National Agricultural Library administered by the Agricultural Research Service (https://data.nal.usda.gov/dataset/data-weather-snow-and-streamflow-data-four-western-juniper-dominated-experimental-catchments, doi:10.15482/USDA.ADC/1254010).

**1 Introduction**

Across the interior western US, native Western Juniper (*Juniperus occidentalis* Hook.) is encroaching into sagebrush-dominated (*Artemisia* spp.) landscapes.  These fire-sensitive native conifers in the western U.S. have greatly expanded in response to changing fire regimes (increased woody fuels in response to fire suppression efforts) following European settlement (**????**). Western Juniper now dominates over 3.6 million ha of rangeland in the Intermountain Western US. Juniper (*Juniperus* spp.) expansion into sagebrush ecosystems influences the vegetation community (**???**) and the hydrology and soil resources of an area (**???**), which  in turn also affects the wildlife habitat. For example, research in similar study sites demonstrate that juniper encroachment diminishes understory biomass (**???**), which serves as a soil stabilization mechanism, forage for livestock, and habitat diversity. At mid to high elevations, expansion of native conifer species is viewed as a major threat to sagebrush obligates such as the greater sage grouse (*Centrocercus*

*urophasianus*) (**??**). Because of the associated impacts on the ecosystem quality and local economy (**?**), juniper encroachment has become a critical issue to the region's resource managers and ranchers.

Although the deleterious impact of juniper encroachment is widely reported through field studies, there are limited datasets available to quantify that impact on larger scales through modeling. To address the need for monitoring data, the South Mountain Experimental Catchments were established in 2007 in a juniper-dominated region of southwestern Idaho, USA (**?**). A period of background data collection spans the 2008-2015 water years. The catchments are now being treated to remove juniper so comparative studies can be conducted. Catchment M was burned in the fall of 2015 and catchment G is scheduled to burn in the spring of 2017. The long term treatment plan includes burning catchments F and then E.

[revised manuscript text omitted]

**4.3 Wind speed and direction**

Wind speed and direction are measured at the six weather stations. The three low elevation sites (M1, F1 & G1) are sheltered by topography and vegetation, while the ridge-top sites (E2, M2 & G2) are wind-exposed. F1 is extremely wind sheltered by both topography and vegetation with a mean wind speed of 0.7 m s$^{-1}$, while M2 is the the most wind-exposed with a mean wind speed of 2.4 m s$^{-1}$. The prevailing wind direction during precipitation is from the west (274°). The maximum wind speed recorded during six water years of the dataset  was 14.3 m s$^{-1}$. We did not attempt to gap fill missing or bad data from the wind direction time series, as correlations between wind measurement stations are low. However, there is sufficient wind

direction data to obtain average wind directions during water years and individual storms. An example of monthly mean wind speed and the monthly range of wind speed values from weather station E2 is shown in Figure **??**c.

**4.4 Incoming solar radiation**

Incoming solar radiation is measured at the six weather stations. Solar radiation measurements from weather stations F1 and M1 are vegetation-affected in the mornings and evenings. The average solar loading at the F1 site was 12.9 MJ day$^{-1}$ m$^{-2}$, while at site E2 it was 16.1 MJ day$^{-1}$ m$^{-2}$. An example of monthly solar loadings from weather station E2 is shown in Figure **??**c.

**5 Snow and streamflow data**

**5.1 Snow data**

Snow depth is continuously measured at the six weather stations. Because these automated snow depth measurements are inherently noisy, the data are processed using multiple smoothing windows. This practice allows for the cleaning of instrument noise, while maintaining sharp accumulation and melt events. We did not attempt to fill large time periods with excessively noisy data in the cleaned snow depth data file, and have denoted them as missing data (Fig. **??**b, WY2011). Excessively noisy data was identified as time periods that contained more erroneous measurements than reasonable measurements. Raw snow depth data are provided. In addition to automated snow depth measurements, manual measurements of snow water equivalent (SWE) were made two to three times each year at snow courses near the six weather stations using a federal-type snow tube. Snow courses were visited 16 times during the  six-year dataset, and were not measured in WY2009. These snow water equivalent values are reported in the final data file, and depths and densities are reported in the raw data file. Although significant resources were expended collecting SWE data, we recognize that this is a limited model validation dataset. The combination of continuous snow depth and SWE measurements should be sufficient to evaluate distributed snow model results. An example of the cleaned snow depth from weather station E2 is shown in Figure **??**b.

**5.2 Streamflow**

Stream stage is measured with Druck pressure transducers in stilling wells at four drop box v-notch weirs. Stage is converted to discharge with well-established rating curves (**?**). The streams that drain the South Mountain Experimental Catchments are intermittent and initiate in response to rain on snow or snow melt events (Fig. **??**a and b). Streamflow ceases in late spring to mid summer. Mean water year discharge from all catchments across years was 115 mm (Fig. **??**). Catchment M, which has the smallest contributing area has the lowest mean annual discharge at 90 mm. Catchment F has the highest mean annual stream discharge at 145 mm. The lowest stream yields occurred in WY2013 and the highest stream yields occurred in WY2011. Runoff ratios are approximated for the four South Mountain Experimental Catchments by assuming that the mean of precipitation measured by gauges within each catchment represents the precipitation that fell in that catchment. Average

catchment runoff ratios varied from 0.07 for M to 0.22 for F. An example of the streamflow data included in this dataset from weir E is shown in Figure **??**b.

**6 Example Data**

150 We present data from a mid February storm in 2012 as an example of the dynamic weather that is described in this manuscript (Fig. **??**). At the start of this storm, the air was cold and saturated resulting in snowfall and an accumulation of about 9.5 cm of snow depth (Fig. **??**b and c). Wind was relatively calm and cloud cover lead to low incoming solar radiation at all weather stations (Fig. **??**a). Snow depth increased until midday on 21 February when air and dew point temperatures rose above freezing and caused precipitation to change from snow to mixed precipitation, and then to rain (Fig. **??**b and c). Snow melt, and rain

155 lead to streamflow initiation from catchments F, G, and M, and an increase in flow at weir E (Fig. **??**c). An additional rain on snow event occurred from 22 February, 17:00 to 19:00 leading to increased streamflow at all weirs. Clear skies and warming temperatures caused increased flow from the smallest  Catchment M on 24 February. A small snow event occurred in the early morning of 25 February, which lead to an increase in snow depth.

**7 Data Availability**

160 All data presented in this paper are available from the National Agricultural Library website (doi:10.15482/USDA.ADC/1254010). Included is a readme file that contains a detailed description of data file contents, including header information and contact information for additional details. Additional weather and hydrologic response data for the South Mountain Experimental Catchments  are available at ftp://ftp.nwrc.ars.usda.gov/publicdatabase/.

**8 Summary**

165 Data presented in this paper support ongoing research in a mountain environment that is relevant to both native ecosystems and local economy in the Great Basin region of the northwestern US. This region has experienced extensive woodland encroachment into sagebrush-dominated landscapes, which has become a critical issue regarding the regional economy and ecosystem health. This publication provides details on background data from catchments that are now juniper dominated. A treatment schedule to remove juniper is now being implemented so comparative studies can be conducted. Catchment M was burned in

[revised manuscript text omitted]